# The Relationship between Magnetic Resonance Imaging and Functional Tests Assessment in Patients with Lumbar Disk Hernia

**DOI:** 10.3390/healthcare11192669

**Published:** 2023-10-01

**Authors:** Bogdan-Alexandru Antohe, Hüseyin Şahin Uysal, Adelina-Elena Panaet, George-Sebastian Iacob, Marinela Rață

**Affiliations:** 1Faculty of Movement, Sports and Health Science, “Vasile Alecsandri” University of Bacău, 600115 Bacău, Romania; antohe.bogdan@ub.ro (B.-A.A.); panaet.adelina@ub.ro (A.-E.P.); 2Faculty of Sport Science, Burdur Mehmet Akif Ersoy University, 15500 Burdur, Turkey; hsuysal@mehmetakif.edu.tr; 3Faculty of Physical Education and Sports, “Alexandru Ioan Cuza” Univesity of Iasi, 700554 Iasi, Romania; georgesebastianiacob@gmail.com

**Keywords:** magnetic resonance imaging, disk hernia, low back pain, functional tests, cross-sectional

## Abstract

Although magnetic resonance imaging (MRI) findings are the gold standard for diagnosing herniated discs, there are many limitations to accessing MRI scanning devices in practice. This study aimed to evaluate the relationship between functional tests (the visual analog scale (VAS), the SLUMP test, the Sciatica Bothersomeness Index (SBI), the Oswestry Disability Index (ODI), and the LASEGUE test and MRI findings (LSA, IVDH L4-L5, IVDH L5-S1, DHS L4-L5, and DHS L5-S1) in patients diagnosed with disc herniation. Seventy-eight patients who met the inclusion criteria participated in the study. Radiologists and neurologists evaluated patients with disc herniation. After the disc hernia diagnosis, the patients were referred to a physical therapist for conservative management of the disk hernia. The physical therapists assessed the pain level and performed functional tests on patients. All statistical analyses were performed using R (Core Team) software. The correlation between the measured variables was conducted using the Pearson and Spearman tests. The study results indicated statistically significant correlations between DHS L4-L5 vertebral level and functional tests (VAS: *r* = 0.49, *p* = 0.00; SBI: *r* = 0.44, *p* = 0.00; ODI: *r* = 0.49, *p* = 0.00; LASEGUE: *r* = −0.48, *p* = 0.00; SLUMP: *r* = 0.50, *p* = 0.00). In conclusion, physiotherapists may prefer functional tests to diagnose the herniated disc, and these functional tests may contribute to performing evidence-based assessments.

## 1. Introduction

Lumbar spine pathology is one of the most important and frequently encountered therapeutic situations worldwide. The primary clinical manifestation is described as localized pain below the rib margins and above the iliac crest, with or without pain radiating in the lower extremity [1]. Approximately 60–80% of the world’s population experiences at least one acute or chronic episode of low back pain (LBP) due to various causes during their lifetime [2,3]. Approximately 39% of the pain is attributed to discogenic reasons, while disk herniations (DH) cause less than 30% [4]. Various environmental and physiological factors can influence DH. Researchers indicated that DH is associated with aging, environmental factors, being overweight, job demands, smoking, and insufficient physical activity [5]. Previous studies reported that DH is the most common diagnosis among degenerative lumbar spine pathologies, with an incidence rate of 2–3% among the global population [6,7]. In addition, the prevalence of the disease is 4.8% for male subjects and 2.5% for women, especially those over 35 years old [4,8].

The diagnosis of DH is primarily made through magnetic resonance imaging (MRI) findings. The interpretation of these findings assists neurologists or physiotherapists in determining the most appropriate intervention procedure (e.g., conservative or surgical) based on the patient’s specific needs [9]. Imaging investigations are often a fundamental part of the clinical diagnosis, and MRI is considered the gold standard for diagnosing DH. The diagnosis of DH can also be made through clinical inspection in addition to a MRI examination. Since therapists may not have access to MRIs in clinical practice, they can use FT to diagnose DH, like LASEGUE, SLUMP, Piriformis Test, and Crossed Straight Leg Test [10,11]. These tests aim to replicate the signs and symptoms of lumbosacral nerve root compression, sciatic nerve irritation, and lower extremity sciatica [10,11].

Although, from a theoretical point of view, clinical testing provides valuable information in the evaluation process, the sensitivity and specificity for many of them may not be not optimal. Because the accuracy of the diagnosis is essential, it is necessary to develop an adapted diagnostic model that emphasizes both the MRI and the clinical examination based on FT. Even if the MRI examination is the best method for DH diagnosis [12], it has limitations in terms of device accessibility, cost, experience, relevance to different populations, and data interpretation [13,14,15]. FT may be an alternative to MRI imaging for field experts with these limitations. This study aimed to evaluate the relationship between functional tests (visual analog scale (VAS), SLUMP test, Sciatica Bothersomeness Index (SBI), Oswestry Disability Index (ODI), and LASEGUE test) and MRI findings (LSA, IVDH L4-L5, IVDH L5-S1, DHS L4-L5, and DHS L5-S1), in patients diagnosed with disc herniation, in order to identify potential correlations between identified variables.

## 2. Materials and Methods

### 2.1. Study Design

This study was designed using a quasi-experimental design according to the correlational research model, one of the quantitative research methods. The study reported the level and direction of the relationship between the MRI findings of the participants and the FT assessment. The Quality Output Checklist and Content Assessment (QuOCCA) checklist was used to improve the methodological quality of the study [16]. Additionally, the details of the study files are presented via the Open Science Framework (OSF) (https://osf.io/rw6zp/ accessed on 11 May 2023). This study was approved by the Vasile Alecsandri University Ethics Committee (no:13/2; 11 August 2023) and was conducted according to the Declaration of Helsinki. Informed consent was obtained from all subjects involved in the study.

### 2.2. Participants

Eighty-one patients (male = 50, female = 31) diagnosed with DH, who were referred to the physical therapy office PlusMed in Bacău, participated in the study. Subjects were identified according to the following criteria: (i) above 18 years of age, (ii) subjects with a MRI-confirmed DH, and (iii) radiating pain or paresis below the hip level. On the other hand, participants with the following criteria were excluded from the study: (i) subjects younger than 18 years of age, (ii) neurosurgery recommendation, spine fractures, previous DH surgery, and (iii) all other diseases that could affect the results of the study.

Participants included in the study were determined according to the purposive sampling method, also known as homogeneous group sampling. Previous research was examined to determine the sample size of the study [17]. Previous research evaluated the relationship between index tests and MRI findings in patients with lumbar disc herniation, and a moderate correlation was reported [17]. Therefore, it was assumed that the study would have a moderate correlation between FT and MRI findings. A priori power analysis was performed (G*Power software, version 3.1, University of Dusseldorf, Dusseldorf, Germany) using the following reference values: bivariate normal model test, two-tailed hypothesis, α = 0.05, power (*β*) = 0.80, and *r* = 0.31. The power analysis result reported that the minimum sample size for this study should be 79 patients. As a result, 81 patients were included in the study, and the analysis results were reported at least 80% power.

### 2.3. Procedures

The MRI images were obtained from Bacău Emergency Hospital and two private hospitals. The MRI images were taken and interpreted by different radiologists. All the participants included in the study had a diagnosis of DH. Since multiple radiologists analyzed the images, we refrained from further classifying the DH due to the potential for differing results. On the other hand, the same experienced therapists performed FT. Patients had no limitations before the test procedures as they were evaluated during their first appointment. The tests were performed in a supine (LASEGUE) or sitting position (SLUMP). The patients were instructed to stay relaxed and cooperate with the physical therapist’s instructions. For enhanced data collection and analysis, the ODI and SBI questionnaires were completed online using Google Forms before conducting the SLUMP and LASEGUE tests.

### 2.4. Measurements

#### 2.4.1. Functional Tests Assessment

##### Crossed Straight Leg Test or LASEGUE’s Sign

LASEGUE’s sign is used to assess the lumbar nerve root compression. The test sensitivity is situated between 0.36 to 0.52, while specificity varies between 0.74 to 0.89 [18,19]. The test was performed by the physical therapist with the patient supine, on the lower limb, with the radiating pain. The patient slowly lifted their leg from the posterior side of the heel, with the knee straight, until the symptoms appeared. The test was noted positive if the movement reproduces the patient’s symptoms. A goniometer was used to measure the hip’s range of motion (ROM) at which the test becomes positive.

##### SLUMP Test

SLUMP test is a neurodynamic test to assess whether a disk hernia or other neural tension contributes to patient symptoms. The test has a sensitivity between 0.44 and 0.84 and a specificity between 0.58 and 0.83 [19,20]. To perform the test, the patient is seated at the table’s edge. The patient was instructed to bend forward their lumbar and thoracic spine and then asked to perform head flexion. If this position does not provoke symptoms, the patient was asked to perform knee extension, followed by ankle dorsiflexion. If the movement reproduces the patient’s symptoms, the test is positive. The test result was noted as negative (0) or positive (1).

##### Sciatica Bothersomeness Index (SBI)

Sciatica Bothersomeness Index is an index in which patients report symptoms during sciatica episodes. The index reports the symptoms’ intensity for the following items: leg pain; numbness or tingling in the leg, foot, or groin; weakness in the leg or foot; back or leg pain while sitting. Each symptom item is rated on a scale from 0 to 6, with anchors at 0 (not bothersome), 3 (somewhat annoying), and 6 (extremely irritating). The total possible score is 24, with higher scores indicating worse symptoms. Patients were instructed to rate the severity of symptoms one week before participating in the study [21].

##### Oswestry Disability Index (ODI)

Oswestry Disability Index is a tool that helps evaluators measure the level of disability in LBP patients [22]. The questionnaire includes the following ten criteria for daily activities: pain intensity, personal care, lifting, walking, sitting, standing, sleeping, sex (if applicable), social, and travel. The score for each item varies from 0 (no disability) to 5 (totally disabled). The final score was obtained by dividing the total possible score (50) by the total score (e.g., 15) and multiplying by 100. Results were reported according to the following reference values: minimal disability (0–20%), moderate disability (20–40%), severe disability (40–60%), crippled (60–80%), and bedbound (80–100%).

##### Visual Analogue Scale (VAS)

The visual analog scale is one of the most widely used methods for pain investigation. Within the VAS scale evaluation, the patient was asked to rate pain intensity, starting from 0 (no pain) to 10 (maximum pain).

#### 2.4.2. Assessment of MRI Images

MRI images were viewed using the RadiAnt DICOM Viewer software (Version 2022.1.1). The images were evaluated in the sagittal plane using T2-weighted images. From the MRI evaluation, the following data were extracted: (i) lumbar spine angle (LSA), (ii) size of the disk herniation (DHS), and (iii) height of the intervertebral disk (IVDH). The size of the herniated disk, the intervertebral disk height, and the lumbar spine angle measurements were performed using RadiAnt software, 2022.1.1, Poznan, Poland. We selected “measurements and tools” and then chose either “length” or “angle.” The measurements were based on millimeters (mm) and followed the protocols recommended by researchers [23].

The size of the herniated disk was measured from a vertical line drawn at the posterior side of the intervertebral disk. From that point, a horizontal line was drawn at the maximum extrusion level (Figure 1). The height of the intervertebral disk was measured between two horizontal lines. One line was drawn on the inferior endplate of the L4 vertebra, and the other was drawn on the superior endplate of the L5 vertebra. The middle of those two lines marked a vertical line, representing the height of the intervertebral disk (Figure 1A). The angle of the lumbar spine was measured by drawing a tangent line on the superior endplate of the L1 vertebra and a second line on the inferior endplate of the L5 vertebra. The lordosis of the lumbar spine was assessed at the intersection of the two lines (Figure 1B).

### 2.5. Statistical Analysis

This study evaluated the relationship between five functional tests and five MRI findings. While three potential moderators (i.e., age, sex, and physical activity level) were identified for the study, 200 correlation analyses were performed with seven subgroups. The study data were reported using descriptive statistics (mean ± standard deviation). For the age variable, two categories were created: (i) 20–40 years old and (ii) 40+ years old. Physical activity level was divided into three categories: (i) sedentary, (ii) moderately active, and (iii) active. The normality distribution was checked using Kolmogorov–Smirnov analysis. The results showed that all study variables met the normality assumption except for the SLUMP functional pain assessment test. Details of the normality analysis are provided in Appendix A.

Pearson Correlation Coefficient (*r*) analysis was performed to analyze the variables that met the normality assumption, while the Spearman Correlation Coefficient analysis was used for the SLUMP variable, which did not have a normal distribution. The correlation coefficient was interpreted based on the following reference values: insignificant (<0.10), small (0.10–0.29), moderate (0.30–0.49), strong (0.50–0.69), very strong (0.70–0.89), or excellent (>0.90) [24]. Statistical analyses were performed using R software (R Core Team). The {ggplot2}, {patchwork}, and {stats} packages were used for statistical analysis and data visualization. The R codes created are presented in Appendix B. All appendices are provided access via OSF (https://osf.io/rw6zp/). All analyses were calculated with a 95% confidence interval, and the statistical significance level was set at *p* < 0.05.

## 3. Results

In this study, data were obtained from a total of 81 participants. Because the data of three participants were incomplete, a total of 78 subjects were included in the analysis (Appendix C). The correlation between the visual analog scale (VAS) and MRI findings was evaluated. The results demonstrate a significant positive and moderate correlation between the VAS and DHS L4-L5 MRI findings (*r* = 0.41, *p* = 0.00). When the VAS and DHS L4-L5 MRI findings were evaluated by sex, a higher level of correlation was found for women (*r* = 0.54, *p* = 0.00). VAS and DHS L4-L5 MRI findings were evaluated according to physical activity level, and a moderately significant result was determined only for sedentary individuals (*r* = 0.45, *p* = 0.00). There is also a significant high-level correlation between the VAS scores of individuals aged 20–40 and DHS L4-L5 findings (*r* = 0.55, *p* = 0.00). Detailed information about the relationship between the VAS scores and MRI findings is provided in Table 1.

When the relationship between the Sciatica Bothersomeness Index (SBI) and MRI findings was evaluated, results like the VAS scores were obtained. A significant and positive correlation was found between SBI and DHS L4-L5 findings (*r* = 0.44, *p* = 0.00). Women had a higher correlation (*r* = 0.49) than men (*r* = 0.42). Regarding physical activity level, significant results were obtained between SBI values of sedentary and moderately active individuals and DHS L4-L5 findings (sedentary individuals: *p* = 0.00; moderately active individuals: *p* = 0.01). In addition, a more significant positive correlation was found between the SBI and DHS L4-L5 findings of younger participants (*r* = 0.57, *p* = 0.00). Detailed information is provided in Table 2.

Like other functional tests, a significant positive moderate-level correlation was found between the Oswestry Disability Index (ODI) and DHS L4-L5 (*r* = 0.49, *p* = 0.00). These correlations were higher in women (*r* = 0.56, *p* = 0.00) than men (*r* = 0.42, *p* = 0.00). The results also found a significant and positive correlation between sedentary and moderately active individuals (sedentary: *r* = 0.50, *p* = 0.03; moderately: *r* = 0.50, *p* = 0.00). Significant results were found between the ODI scores and DHS L4-L5 findings in individuals between 20 and 40 years of age (*p* = 0.00) and individuals over 40 years of age (*p* = 0.00). Detailed information between the ODI scores and MRI findings is provided in Table 3.

When the relationship between the LASEGUE tests and the MRI findings was evaluated, only a significant difference was found between DHS L4-L5 findings (*p* = 0.00). Contrary to other functional tests, a moderate negative correlation was identified between the two variables (*r* = 0.48). A strong negative correlation was observed between men’s LASEGUE (grades) scores and DHS L4-L5 findings (*r* = −0.53, *p* = 0.00). Regarding physical activity level, it was determined that there was a negative and strong-level significant correlation between LASEGUE test scores of moderately active individuals and DHS L4-L5 findings (*r* = −0.61, *p* = 0.00). No statistically significant difference existed between LASEGUE test scores and DHS L4-L5 findings of individuals who actively participate in sports (*r* = −0.08, *p* = 0.86). A more excellent negative correlation was found between younger individuals’ LASEGUE test scores and DHS L4-L5 findings (*r* = −0.59, *p* = 0.00). The relationship between the LASEGUE test scores and MRI findings is provided in Table 4.

The relationship between the SLUMP scores and MRI findings was evaluated, and a significant positive correlation was found only between DHS L4-L5 findings (*r* = 0.50, *p* = 0.00). The relationship between men’s SLUMP scores and DHS L4-L5 findings was more significant (*r* = 0.50, *p* = 0.00). Significant differences were identified between the SLUMP and DHS L4-L5 findings of only sedentary individuals regarding physical activity level (*r* = 0.53, *p* = 0.00). Significant differences were observed between the SLUMP scores and DHS L4-L5 findings for age categories (individuals aged 20–40: *p* = 0.00; individuals over 40: *p* = 0.00). A more significant positive correlation was discovered between the SLUMP scores and DHS L4-L5 findings in younger participants (*r* = 0.57, *p* = 0.00). The relationship between the SLUMP scores and MRI findings is provided in Table 5. The relationship between functional tests and MRI findings was summarized with a dot graph, and details about the results are presented in Figure 2.

## 4. Discussion

The study results indicated a statistically significant correlation between the FT and the MRI findings at the L4-L5 vertebral level. Independent variables, such as sex, age, and physical activity status, influenced our findings. After analyzing the correlations between the VAS and MRI findings, a moderate correlation (*r* = 0.41) was found between pain and the size of the disc hernia at the L4-L5 intervertebral level. The other variables that showed a strong level of correlation were sex (female, *r* = 0.54), physical activity level (sedentary people, *r* = 0.45), and age (20–40 years old category, *r* = 0.55). The L5-S1 vertebral level did not correlate with the pain and the variables we measured.

According to the results, age is a significant risk factor for intervertebral back pain and intervertebral disc pathology. The literature reports a higher incidence of intervertebral disc pathology in individuals younger than 50 [25], with more severe episodes of LBP in individuals aged between 25 and 44 [26]. The higher water content in the intervertebral disk (IVD) among younger individuals may account for a more pronounced experience of pain in an IVD hernia at the L4-L5 vertebral level [27]. This idea is supported by studies in the literature, which highlight an inverse relationship between age and water content of the IVD [27,28]. Therefore, the higher the water content of the IVD, the greater the DH and pain will be. Also, researchers reported that apart from DH, many secondary spinal diseases can lead to symptoms such as spondylolisthesis and facet joint arthritis [29]. So, in our study, LBP may not be solely related to IVD pathology.

The size of the DH correlates with all the dependent variables that we measured (VAS, ODI, SBI, LASEGUE, and SLUMP) at the L4-L5 vertebral level. Our findings were consistent with a previous study that found degeneration of the L4-L5 IVD was degenerated in 41.2% of individuals [30]. At the L5-S1 vertebral level, there was no correlation between the size of the disk hernia and the variables we measured. Since we have achieved statistical significance at the L4-L5 IVDH, it can be said that there is a 50% correlation between the FT and MRI findings. This study found an agreement between clinical and radiological findings in 69% of patients diagnosed with DH, involving 105 subjects [20]. The average values of our patients with IDH were 3.5 mm for the L4-L5 vertebra and 3.44 mm for the L5-S1 vertebra. The values of L5-S1 vertebral DH are more minor, measuring 0.06 mm, compared to the L4-L5 vertebral level. This discrepancy is insufficient to warrant the distinction between the correlation coefficients. Our study, as well as those in the literature, is debating the role of DH in patients’ symptoms. Some studies demonstrate a relationship between the MRI findings and LBP [31], as well as IVD degeneration and the patient’s symptoms [32]. In contrast to this supposition, some researchers have shown no significant correlation between the compromised area of the spinal canal [33] or the root compression observed in MRI and clinical symptoms [34]. Since the frequency of small hernias (less than 5 mm) is high in asymptomatic subjects, we can consider various physiological mechanisms that may be causing sciatica symptoms (aside from mechanical pain), such as biochemical factors [35] or other variables related to pain (physical activity, type of work, posture, etc.).

In terms of physical activity level, this is one of the most critical factors that influenced the results of our research. At the L4-L5 vertebral level, the subjects’ physical activity influenced the level of pain (sedentary, *r* = 0.41), level of bothersomeness (sedentary, *r* = 0.42; average active, *r* = 0.55), ODI (sedentary, r = 0.50; average active, *r* = 0.50), LASEGUE (sedentary, r = 0.48; average active, *r* = 0.61), and SLUMP (sedentary, *r* = 0.53). The physical activity level was also strongly correlated with the SBI (*r* = 0.76) and the LASEGUE test (*r* = 0.87) at the L5-S1 vertebral level in active individuals. In contrast to our other tested variables, this variable also showed a very strong correlation with the L5-S1 level. Considering our findings, the level of physical activity influences the vertebral levels L4-L5 and L5-S1 differently. That means the root of the problem could be different in active individuals compared to sedentary individuals. Since the subjects in the active category had physically demanding jobs, the cause of LBP could be biomechanical, resulting from overloading the spine. The pain mechanism could be related to postural deficits in the L4-L5 vertebral level, where people are sedentary. Therefore, our results confirm previous arguments in the literature, which suggest that the risks of LBP and sciatic pain are significantly influenced by occupation, particularly in individuals with sedentary lifestyles [36]. Also, engaging in moderate physical activity [37] or participating in sports activities was associated with a lower prevalence of chronic LBP [38].

The SBI showed a moderate correlation with the size of L4-L5 disk hernia in both women (*r* = 0.49) and men (*r* = 0.42), with a stronger correlation observed in younger subjects (*r* = 0.57). Our results agree with the research that states that bothersomeness is 10% higher in women [21]. On the other hand, we did not find any other correlations between MRI characteristics and SBI values. After conducting the study and reviewing the literature, we can conclude that this type of assessment can be utilized to predict improvements in the physical health of the subject following a treatment (either conservative or surgical). Still, it is not suitable for diagnosis [39,40]. It appears that there is a significant negative correlation between the lumbar spine angle and SBI values (*r* = −0.59), ODI (*r* = −0.53), and SLUMP test (*r* = −0.54) in the average active population. As the lumbar lordosis decreases, the symptoms of the patient increase. This data supports the literature’s viewpoint that sagittal alignment can result in load-bearing and force-distribution changes in the lumbar spine [41], leading to LBP. The ODI questionnaire had a moderate correlation with the size of the L4-L5 disk hernia (*r* = 0.49). Women (*r* = 0.56) had stronger correlations with disk hernia size than men (*r* = 0.42). Also, the ODI values were statistically significant only in sedentary (*r* = 0.50) and moderately active (*r* = 0.50) individuals.

Reporting our results to the literature, the data are debated. Some studies say the ODI scores were not statistically significantly affected by spondylolisthesis, multilevel disease, or the degree of stenosis [42]. Other studies agree that increased lumbar intervertebral disk disease in MRI goes along with an increased ODI [43], and it can identify the worst diagnosis, like lumbar spine stenosis [44]. Since the data in the literature is debated and our level of correlation is not very strong, we believe the ODI questionnaire is comprehensive (because it analyzes many daily activities). Still, it cannot play an essential role in clinical diagnosis. As in the case of SBI, we believe the ODI can be a valuable tool to measure a patient’s outcome.

LASEGUE’s test has a moderately inverse correlation with the DH at the L4-L5 level (*r* = −0.48). The test has a stronger correlation in men (*r* = −0.53) than women (*r* = −0.44). Similar results were found in a study where men had 1.3 times more chances of a positive LASEGUE than women [45]. The same pattern is found in the SLUMP test, where males correlate more with disk hernia at the L4-L5 vertebral level (0.55) than females (*r* = 0.44). The higher values registered by men in the two functional tests indirectly indicate that they are more susceptible to disc herniations. The LASEGUE test shows a strong correlation with L4-L5 disk hernia in the 20–40 age category (*r* = −0.59), compared to the 40+ age category (*r* = −0.42). Our results are similar to those obtained by the study that states LASEGUE’s test is positive in younger individuals [18]. As a patient ages, the test’s ability to diagnose positive sciatica patients diminishes. If we compare the SLUMP test and LASEGUE’s test, the SLUMP test showed a stronger correlation value (*r* = 0.50) than LASEGUE’s (*r* = −0.48). In the literature, the SLUMP test has a higher sensitivity (55.3%) than the LASEGUE test (18.1%). The SLUMP test has a better outcome because head flexion creates greater tension on the sciatic nerve [46,47]. An important aspect is that the LASEGUE test has a very strong correlation at the level of L5-S1 disk hernia in active people (*r* = −0.87). This category comprises heavy workers, and their DH could be caused by mechanical overloading. These results oppose the other situation where people with less physical activity tend to have DH. According to our results and the evidence from the literature, we can conclude that the gold standard for DH diagnosis is SLUMP and not the LASEGUE test [19,48].

### 4.1. Limitations

In our study, there are some limitations. Firstly, we did not measure other critical, independent variables that could have influenced the results of our study. These factors include weight, height, BMI, education level, smoking, and strength and flexibility of the lumbar spine. We believe all these factors could provide a better understanding of our results. Another limitation is the fact that we did not classify the disk hernia. We did not type the disk hernia because different specialists interpreted the results of the MRI, and the results could differ.

### 4.2. Conclusions

Physical activity level was the most critical risk factor for subjects diagnosed with disk hernia. Study results indicated that active people are more protected from herniated disks. This study confirmed the SLUMP test as the gold standard for diagnosing a herniated disc. In our subjects, we found stronger correlations between functional tests and L4-L5 vertebral level. Disk herniation has a stronger manifestation in younger people. After analyzing all the results and comparing them with the literature, it was strongly identified that clinical assessment is an essential tool for patient diagnosis. However, specialists must associate the results with MRI or other investigation findings. It can also be said that SLUMP and LASEGUE’s tests have more clinical diagnostic value than the ODI and SBI, which are more closely related to patient outcomes.

## Figures and Tables

**Figure 1 healthcare-11-02669-f001:**
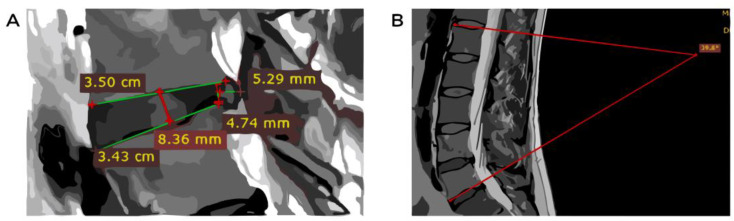
Intervertebral disk hernia measurements. (**A**) Intervertebral disk hernia and height measurements. (**B**) Lumbar spine angle measurements.

**Figure 2 healthcare-11-02669-f002:**
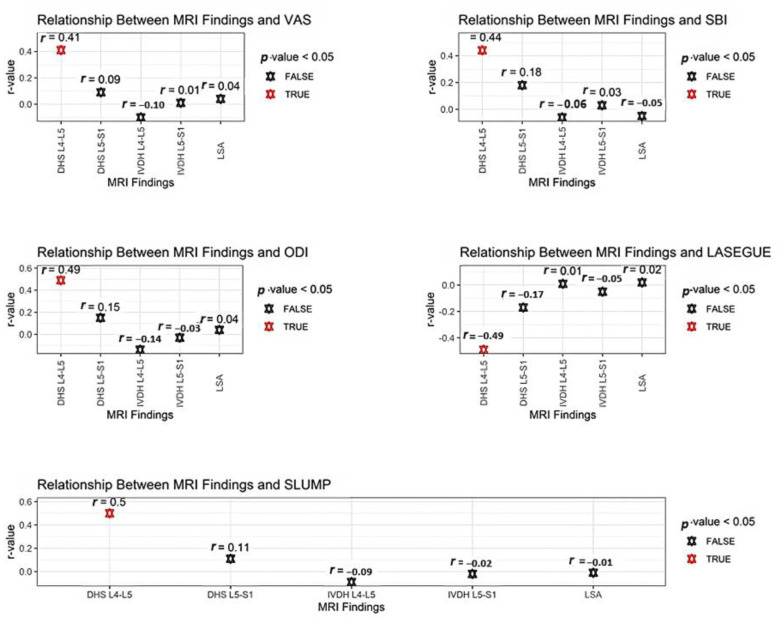
The relationship between functional tests and MRI findings.

**Table 1 healthcare-11-02669-t001:** Results of the relationship between visual analog scale and MRI findings.

MRI Finding	VAS *r* (*p*-Value)	Sex r (*p*-Value)		Physical Activity Level r (*p*-Value)			Age r (*p*-Value)	
		Male (n = 48)	Female (*n* = 30)	Sedentary (*n* = 52)	Moderate Active (*n* =18)	Active (*n* = 8)	20–40 Age (*n* = 31)	Above 40 Age (*n* = 47)
LSA	0.04 (0.68)	0.15 (0.29)	−0.05 (0.77)	0.21 (0.13)	−0.37 (0.13)	−0.40 (0.31)	0.08 (0.66)	−0.00 (0.99)
IVDH L4-L5 (mm)	−0.10 (0.35)	−0.22 (0.13)	0.06 (0.72)	−0.08 (0.56)	−0.08 (0.74)	−0.43 (0.28)	−0.21 (0.23)	−0.02 (0.86)
IVDH L5-S1 (mm)	0.01 (0.89)	−0.01 (0.93)	0.09 (0.62)	−0.04 (0.73)	0.21 (0.38)	−0.04 (0.90)	−0.02 (0.87)	0.04 (0.78)
DHS L4-L5 (mm)	0.41 (0.00) *	0.32 (0.02) *	0.54 (0.00) *	0.45 (0.00) *	0.37 (0.12)	0.20 (0.62)	0.55 (0.00) *	0.28 (0.05)
DHS L5-S1 (mm)	0.09 (0.42)	0.10 (0.48)	0.06 (0.74)	0.07 (0.62)	0.03 (0.89)	0.50 (0.20)	0.01 (0.94)	0.15 (0.31)

Note. LSA: lumbar spine angle; IVDH: intervertebral disk height (mm); DHS: disk hernia size; VAS: visual analog scale. * There was a statistically significant difference. These analyses were performed using the Pearson Correlation Coefficient.

**Table 2 healthcare-11-02669-t002:** Results of the relationship between Sciatica Bothersomeness Index and MRI findings.

MRI Finding	SBI *r* (*p*-Value)	Sex *r* (*p*-Value)		Physical Activity Level *r* (*p*-Value)			Age *r* (*p*-Value)	
		Male (*n* = 48)	Female (*n* = 30)	Sedentary (*n* = 52)	Moderate Active (*n* =18)	Active (*n* = 8)	20–40 Age (*n* = 31)	Above 40 Age (*n* = 47)
LSA	−0.05 (0.65)	0.00 (0.95)	−0.10 (0.58)	0.12 (0.36)	−0.59 (0.00) *	−0.54 (0.15)	−0.05 (0.77)	−0.11 (0.45)
IVDH L4-L5 (mm)	−0.06 (0.57)	−0.15 (0.28)	0.08 (0.67)	−0.05 (0.70)	−0.01 (0.95)	−0.14 (0.73)	−0.09 (0.59)	−0.06 (0.68)
IVDH L5-S1 (mm)	0.03 (0.76)	−0.00 (0.96)	0.16 (0.37)	−0.03 (0.82)	0.34 (0.16)	−0.18 (0.65)	0.06 (0.71)	0.00 (0.95)
DHS L4-L5 (mm)	0.44 (0.00) *	0.42 (0.00) *	0.49 (0.00) *	0.42 (0.00) *	0.55 (0.01) *	0.21 (0.60)	0.57 (0.00) *	0.34 (0.01) *
DHS L5-S1 (mm)	0.18 (0.10)	0.18 (0.20)	0.15 (0.40)	0.15 (0.25)	0.06 (0.78)	0.76 (0.02) *	0.03 (0.84)	0.27 (0.06)

Note. LSA: lumbar spine angle; IVDH: intervertebral disk height (mm); DHS: disk hernia size; SBI: Sciatica Bothersomeness Index. * There was a statistically significant difference. These analyses were performed using the Pearson Correlation Coefficient.

**Table 3 healthcare-11-02669-t003:** Results of the relationship between Oswestry Disability Index and MRI findings.

MRI Finding	ODI *r* (*p*-Value)	Sex *r* (*p*-Value)		Physical Activity Level *r* (*p*-Value)			Age *r* (*p*-Value)	
		Male (*n* = 48)	Female (*n* = 30)	Sedentary (*n* = 52)	Moderate Active (*n* =18)	Active (*n* = 8)	20–40 Age (*n* = 31)	Above 40 Age (*n* = 47)
LSA	0.04 (0.67)	0.20 (0.15)	−0.11 (0.55)	0.22 (0.11)	−0.53 (0.02) *	−0.47 (0.23)	0.01 (0.92)	0.02 (0.86)
IVDH L4-L5 (mm)	−0.14 (0.19)	−0.18 (0.21)	−0.10 (0.59)	−0.12 (0.37)	−0.15 (0.52)	−0.33 (0.44)	−0.29 (0.19)	−0.10 (0.50)
IVDH L5-S1 (mm)	−0.03 (0.72)	−0.01 (0.93)	−0.09 (0.60)	−0.08 (0.53)	0.19 (0.43)	0.00 (0.99)	−0.02 (0.90)	−0.05 (0.69)
DHS L4-L5 (mm)	0.49 (0.00) *	0.42 (0.00) *	0.56 (0.00) *	0.50 (0.00) *	0.50 (0.03) *	0.43 (0.28)	0.51 (0.00) *	0.45 (0.00) *
DHS L5-S1 (mm)	0.15 (0.17)	0.13 (0.35)	0.19 (0.30)	0.08 (0.53)	0.13 (0.59)	0.57 (0.13)	0.12 (0.50)	0.16 (0.26)

Note. LSA: lumbar spine angle; IVDH: intervertebral disk height (mm); DHS: disk hernia size; ODI: Oswestry Disability Index. ***** There was a statistically significant difference. These analyses were performed using the Pearson Correlation Coefficient.

**Table 4 healthcare-11-02669-t004:** Results of the relationship between LASEGUE’s test and MRI findings.

MRI Finding	LASEGUE *r* (*p*-Value)	Sex *r* (*p*-Value)		Physical Activity Level *r* (*p*-Value)			Age *r* (*p*-Value)	
		Male (*n* = 48)	Female (*n* = 30)	Sedentary (*n* = 52)	Moderate Active (*n* =18)	Active (*n* = 8)	20–40 Age (*n* = 31)	Above 40 Age (*n* = 47)
LSA	0.02 (0.83)	−0.16 (0.25)	0.15 (0.41)	−0.08 (0.52)	0.34 (0.15)	0.56 (0.14)	−0.08 (0.92)	0.02 (0.88)
IVDH L4-L5 (mm)	0.01 (0.87)	−0.01 (0.90)	0.11 (0.54)	−0.02 (0.85)	0.33 (0.16)	−0.04 (0.91)	0.09 (0.61)	−0.04 (0.78)
IVDH L5-S1 (mm)	−0.05 (0.64)	−0.20 (0.16)	0.15 (0.40)	−0.11 (0.42)	−0.05 (0.82)	0.25 (0.53)	−0.06 (0.73)	−0.05 (0.73)
DHS L4-L5 (mm)	−0.48 (0.00) *	−0.53 (0.00) *	−0.44 (0.02) *	−0.48 (0.00) *	−0.61 (0.00) *	−0.08 (0.86)	−0.59 (0.00) *	−0.42 (0.00) *
DHS L5-S1 (mm)	−0.17 (0.14)	−0.20 (0.20)	−0.12 (0.54)	−0.13 (0.37)	0.00 (0.98)	−0.87 (0.01) *	−0.20 (0.31)	−0.17 (0.26)

Note. LSA: lumbar spine angle; IVDH: intervertebral disk height (mm); DHS: disk hernia size. ***** There was a statistically significant difference. These analyses were performed using the Pearson Correlation Coefficient.

**Table 5 healthcare-11-02669-t005:** Results of the relationship between SLUMP test and MRI findings.

MRI Finding	SLUMP *r* (*p*-Value)	Sex *r* (*p*-Value)		Physical Activity Level *r* (*p*-Value)			Age *r* (*p*-Value)	
		Male (*n* = 48)	Female (*n* = 30)	Sedentary (*n* = 52)	Moderate Active (*n* =18)	Active (*n* = 8)	20–40 Age (*n* = 31)	Above 40 Age (*n* = 47)
LSA	−0.01 (0.92)	0.05 (0.72)	−0.07 (0.67)	0.18 (0.19)	−0.54 (0.01) *	−0.46 (0.24)	0.13 (0.45)	−0.18 (0.20)
IVDH L4-L5 (mm)	−0.09 (0.41)	−0.20 (0.16)	0.09 (0.62)	−0.10 (0.46)	−0.09 (0.72)	0.01 (0.97)	−0.12 (0.50)	−0.09 (0.52)
IVDH L5-S1 (mm)	−0.02 (0.81)	−0.04 (0.78)	0.00 (0.96)	−0.14 (0.30)	0.22 (0.37)	0.29 (0.47)	0.02 (0.90)	−0.06 (0.65)
DHS L4-L5 (mm)	0.50 (0.00) *	0.55 (0.00) *	0.44 (0.01) *	0.53 (0.00) *	0.46 (0.05)	0.43 (0.27)	0.57 (0.00) *	0.43 (0.00) *
DHS L5-S1 (mm)	0.11 (0.32)	0.20 (0.16)	−0.06 (0.72)	0.30 (0.79) *	0.24 (0.32)	0.36 (0.37)	−0.03 (0.87)	0.21 (0.15)

Note. LSA: lumbar spine angle; IVDH: intervertebral disk height (mm); DHS: disk hernia size; SLUMP. ***** There was a statistically significant difference. These analyses were performed using the Spearman Correlation Coefficient.

## Data Availability

Data of this study are presented on OSF **(**https://osf.io/rw6zp/, accessed on 1 March 2023).

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
