# Peer review of "The Relationship between Magnetic Resonance Imaging and Functional Tests Assessment in Patients with Lumbar Disk Hernia"

_healthcare, 2023, doi:10.3390/healthcare11192669_

Round 1

Reviewer 1 Report

First of all, congratulations on the methodology and the writing of the manuscript. It is clear and without errors. On the other hand I am enclosing some slight suggestions for improvement of your great work:

In the INTRODUCTION I suggest including some more current references about risk factors and low back pain. As well as the main clinical guidelines on the subject.

Behennah J, Conway R, Fisher J, Osborne N, Steele J. The relationship between balance performance, lumbar extension strength, trunk extension endurance, and pain in participants with chronic low back pain, and those without. Clin Biomech. 2018;53:22–30. 

Sitthipornvorakul E, Janwantanakul P, Lohsoonthorn V. The effect of daily walking steps on preventing neck and low back pain in sedentary workers: a 1-year prospective cohort study. Eur Spine J. 2015;24(3):417–24.

Shiri R, Karppinen J, Leino-Arjas P, Solovieva S, Viikari-Juntura E. The association between obesity and low back pain: a meta-analysis. Am J Epidemiol. 2010;171(2):135–54.

Alzahrani H, Alshehri M, Attar WA, Alzhrani M. (320) The Association between Sedentary Behavior and Low Back Pain: A Systematic Review and Meta-Analysis of Longitudinal Studies. J Pain. 2019 Apr;20(4):S55. 

Lack of strength and flexibility as well as sedentary lifestyles or lack of physical activity are factors that should have been correlated. If this could not be done now, it should be included as limitations of the study.

Once again, I thank you for your work.

Author Response

Dear reviewer,
We upload a word document with the revisions made, based of your sugesstions. We also modified the manuscript, according with your suggestions.
Thank you!

Reviewer 2 Report

The relationship between magnetic resonance imaging and functional tests assessment in patients with lumbar disk hernia: A cross sectional study

Thank you very much for allowing me to review the current manuscript. It is a topic of interest and relevance in pain management. However, there are certain aspects to consider in order to improve the manuscript:

Abstract:

- The statistical software used should be indicated.

- I believe the term 'cross-sectional' could be included as a keyword.

Introduction:

- The objective of the introduction does not align with what is stated in the abstract. Please unify.

Methods:

- I am concerned that the sample size may be insufficient to achieve adequate statistical power. Please provide further clarification on the achieved statistical power.

- Is Figure 1 an original creation?

- The objective should not be included in the 'Statistical Analysis' section.

Results:

- Were sociodemographic variables measured in the sample? If so, please indicate. It is important to include this information to contextualize the sample characteristics.

- Table 1: Please specify which statistical analyses are presented in the table. What is the meaning of the asterisks?

- Same for Table 2.

- The term 'gender' is not correct. In this case, when distinguishing between males and females, what is being considered is 'sex.' Gender analysis involves taking into account socio-cultural aspects of gender. Please correct this throughout the manuscript.

- The results of the models would have more power if they were adjusted for other related factors, such as BMI, etc. Consider the need to repeat the analyses taking this into account. In the limitations, it has been mentioned that certain variables related to the topic were not measured. Could you provide clarification on whether other relevant variables were recorded?

Discussion-Limitations: Please include a section with the final conclusions.

Author Response

(The authors gave the same response as above.)

Round 2

Reviewer 2 Report

Dear authors,
I believe that the article has improved substantially with the included changes and they have been able to correctly answer all my questions. I would just like to clarify one of them:

They could include in the methodology section the information they add in this response in a clearer way in the manuscript, indicating the models and correlations that have been made.
Author Answer: Dear Reviewer, as we mentioned before, each moderator variable would
cause the scope of the study to expand further. In the current study, we used five tables and one
figure to report the results. We performed a total of 200 correlation analyzes on these tables
(reporting the effect of seven categorical variables on the results in addition to the relationship
of five functional tests to five MRI findings). We increased our number of references to discuss
these results effectively. Even in the current form of the study, the Dear Editor asked us to
reduce the number of references. Your suggestions are very valuable to us and the subject area.
Therefore, we stated in the limitations section that the BMI variable should also be investigated.

Author Response

Dear reviewer,

Thank you for your suggestions. They really help us to improve our manuscript.
We upload the word document with the changes that you required.

Have a great day!
Bogdan,